# Occurrence of L1M Elements in Chromosomal Rearrangements Associated to Chronic Myeloid Leukemia (CML): Insights from Patient-Specific Breakpoints Characterization

**DOI:** 10.3390/genes14071351

**Published:** 2023-06-27

**Authors:** Alberto L’Abbate, Vittoria Moretti, Ester Pungolino, Giovanni Micheloni, Roberto Valli, Annalisa Frattini, Matteo Barcella, Francesco Acquati, Rolland A Reinbold, Lucy Costantino, Fulvio Ferrara, Alessandra Trojani, Mario Ventura, Giovanni Porta, Roberto Cairoli

**Affiliations:** 1Institute of Biomembranes, Bioenergetics, and Molecular Biotechnologies, National Research Council (IBIOM-CNR), 70125 Bari, Italy; a.labbate@ibiom.cnr.it; 2Genomic Medicine Research Center, Department of Medicine and Surgery, University of Insubria, Via JH Dunant 5 Varese, 21100 Varese, Italy; vmoretti@uninsubria.it (V.M.); roberto.valli@uninsubria.it (R.V.); giovanni.porta@uninsubria.it (G.P.); 3Division of Hematology, ASST Grande Ospedale Metropolitano Niguarda, 20162 Milano, Italy; estermaria.pungolino@ospedaleniguarda.it (E.P.); alessandra.trojani@ospedaleniguarda.it (A.T.); roberto.cairoli@ospedaleniguarda.it (R.C.); 4Genetics and Biomedical Research Institute, National Research Council (IRGB-CNR), 20090 Milano, Italy; annalisa.frattini@irgb.cnr.it; 5Department of Health Science, University degli Studi of Milan, Via Rudini 8, 20142 Milan, Italy; matteobarcella84@gmail.com; 6Department of Biotechnology and Life Science, University of Insubria, Via JH Dunant 3, 21100 Varese, Italy; francesco.acquati@uninsubria.it; 7Genomic Medicine Research Center, Department of Biotechnology and Life Science, University of Insubria, Via JH Dunant 3, 21100 Varese, Italy; 8Institute of Biomedical Technologies, National Research Council of Italy, 20054 Segrate, Milano, Italy; rollandreinbold@me.com; 9Department of Molecular Genetics, Centro Diagnostico Italiano, 20147 Milano, Italy; lucy.costantino@cdi.it (L.C.); fulvio.ferrara@cdi.it (F.F.); 10Department of Biology, University of Bari ‘Aldo Moro’, Via Edoardo Orabona 4, 70124 Bari, Italy

**Keywords:** LINE, L1M, chronic myeloid leukemia, rearrangements, BCR-ABL1

## Abstract

Chronic myeloid leukemia (CML) is a rare myeloproliferative disorder caused by the reciprocal translocation t(9;22)(q34;q11) in hematopoietic stem cells (HSCs). This chromosomal translocation results in the formation of an extra-short chromosome 22, called a Philadelphia chromosome (Ph), containing the *BCR-ABL1* fusion gene responsible for the expression of a constitutively active tyrosine kinase that causes uncontrolled growth and replication of leukemic cells. Mechanisms behind the formation of this chromosomal rearrangement are not well known, even if, as observed in tumors, repetitive DNA may be involved as core elements in chromosomal rearrangements. We have participated in the explorative investigations of the PhilosoPhi34 study to evaluate residual Ph+ cells in patients with negative FISH analysis on CD34+/lin- cells with gDNA qPCR. Using targeted next-generation deep sequencing strategies, we analyzed the genomic region around the t(9;22) translocations of 82 CML patients and one CML cell line and assessed the relevance of interspersed repeat elements at breakpoints (BP). We found a statistically higher presence of LINE elements, in particular belonging to the subfamily L1M, in BP cluster regions of both chromosome 22 and 9 compared to the whole human genome. These data suggest that L1M elements could be potential drivers of t(9;22) translocation leading to the generation of the *BCR-ABL1* chimeric gene and the expression of the active *BCR-ABL1*-controlled tyrosine kinase chimeric protein responsible for CML.

## 1. Introduction

Genomic instability is a hallmark of most types of cancer and leads to structural variants (SVs) including translocations, inversions, deletions, and duplications [1]. Genomic changes, ranging from simple translocations to complex karyotypes that arise from tangled phenomena such as chromothripsis, reshape cancer genomes and create de novo fusion genes with oncogenic properties [2]. Genomic rearrangements originate from DNA double strand breaks (DSBs) and misrepair through non-homologous end joining (NHEJ), which requires little or no homology, or homologous recombination (HR) [3].

Repeated DNA sequences are implicated as promoting elements of some of these recurrent chromosomal rearrangements [4]. The high density of repetitive DNA in some BP regions suggests that these sequences may provide “hot spots” for recombination and mediate translocation processes, increasing the likelihood of chromosomal rearrangement phenomena [4]. Four classes of repeated and interspersed sequences have been identified in the mammalian genome: DNA transposons, and LINE (long interspersed nuclear element), SINE (short interspersed nuclear element), and LTR (long terminal repeat) retrotransposons. The last three classes are the main represented categories of retrotransposons in mammals and constitute about 20%, 13%, and 8% of the human genome, respectively [5]. These elements, previously described as “junk DNA”, are now considered key components for the evolution and plasticity of the genome [5]. Their involvement in genetic dysfunctions is now evident, as they can trigger chromosomal rearrangements and cause mutations, resulting in the development of different pathologies including cancer [5]. The LINE’s most common family is L1, while Alu sequences are the most represented SINE elements. Both these elements are classified into subfamilies based on their sequence variants and evolution: L1 can be divided into L1M (mammalian-specific, oldest), L1P (primate-specific, intermediate), and L1H (human-specific, youngest) subfamilies; Alu elements are classified as *Alu*J (oldest), *Alu*S (intermediate), and *Alu*Y (youngest) [6]. The recombination between Alu elements has been reported to be responsible for the partial duplication of the *MLL* gene in acute myeloid leukemia as well as in the generation of reciprocal translocations in tumors [7].

With the development of next generation sequencing (NGS) technologies (whole genome sequencing, exome, and targeted panels), a deeper understanding of genetic mechanisms involved in aberrant genetic alterations has become possible [8]. Limited efforts have been made to clarify molecular phenomena underlying the formation of such non-random genomic rearrangements, and little is known regarding the mechanisms responsible for a variety of translocations [9]. The precise knowledge of sequences surrounding BPs is fundamental to exploring genetic causes in patients with balanced translocations or inversion.

Chronic myeloid leukemia is a clonal myeloproliferative disorder characterized by the reciprocal translocation t(9;22)(q34;q11), involving the proto-oncogene *ABL1* (Abelson leukemia kinase proto-oncogene) on chromosome 9 and the *BCR* (breakpoint cluster region) on chromosome 22. The translocation generates the characteristic Ph chromosome and *BCR-ABL1* fusion gene [10], a constitutively active tyrosine kinase that promotes proliferation and survival of leukemic cells through the activation of downstream pathways such as RAS, RAF, JUN kinase, MYC, and STAT [11].

Three well-defined *BCR* BP regions in chromosome 22 have been characterized: the most frequent, named the Major BP cluster region (M-bcr), occurs between exons 12 and 16. CML patients show mostly M-bcr BPs resulting in a b2a2 or b3a2 transcript, containing *BCR* exon 13 (b2) or 14 (b3) and *ABL1* exon 2 (a2) [12]. In a small subset of patients, the region between exons 19 and 20 (micro-bcr, μ-bcr) or the region distal to exon 1 (minor-bcr, m-bcr) in the *BCR* are involved. The BP region in *ABL1* includes a more variable and extended region of about 200 kb, from 10 kb upstream of the 5′ of the gene to exon 2 [12]. These BP regions are associated with the production of p210, p230, and p190 BCR-ABL fusion protein variants, respectively.

Previously, in a cohort of 27 CML patients, we identified the presence of Alu at the *BCR-ABL1* BP junctions, underlying the evidence that repeated sequences may facilitate the pairing process and the resulting chromosomal translocation [9].

As participants in the exploratory investigations of the PhilosoPhi34 study that investigates the efficacy of nilotinib 300 mg BID in depleting bone marrow (BM) leukemic stem cells (CD34+/linPh+) in newly diagnosed chronic-phase (CP) CML patients at specific time points of treatment [13], we have analyzed a cohort of 82 CML patients with the goal to identify the patient-specific rearrangement of the BCR/ABL1 in order to evaluate the minimal residual disease (MRD) in the Ph^+^ CML patients after 6 or 12 months of nilotinib treatment. The aim of this study was to evaluate, through a gDNA–qPCR assay designed on patient-specific BP sequences, the number of Ph^+^ residual cells [9,14,15,16,17].

We identified patients’ BPs with the target enrichment approach and next generation sequencing (NGS) [14,15,16,17]. Moreover, we have included the KCL22 CML cell line as a positive control in our analyses. The analysis showed that *BCR-ABL1* junctions were clustered in 9 kbs on chromosome 22, mostly including the M-bcr BP cluster region, while on chromosome 9 they were clustered in a wider region of 154 kbs. Thus, we investigated the causes behind the generation of the Ph chromosome by analyzing sequences flanking each single breakpoint.

The analysis of repetitive elements in these regions showed a high presence of LINE elements belonging to the subfamily L1M. The comparison of L1M elements’ frequency at the BP regions compared with their distribution throughout the entire human genome highlighted that L1M elements were much more recurrent at BP regions than in the whole genome. Therefore, we hypothesized, for the first time, the involvement of L1M in the translocation t(9;22)(q34;q11) and in the consequent creation of the Ph chromosome and *BCR-ABL1* fusion gene responsible for CML.

## 2. Materials and Methods

For any specific details of the analysis we performed, look at Appendix B.

### 2.1. Patient Samples

The PhilosoPhi34 study, which included 15 centers in Italy, collected bone marrow (BM) samples from 87 consecutive patients with CML on behalf of the Rete Ematologica Lombarda (REL). The PhilosoPhi34 study enrolled newly diagnosed Ph^+^ CML patients in the chronic phase (CP-CML), aged ≥ 18 years, either male or female. BM samples were obtained and analyzed in accordance with the declaration of Helsinki, after written consent. We analyzed 82 CML patients of the PhilosoPhi34 study.

### 2.2. Selection of BM CD34+/lin- Cells

Mononuclear cells (MNCs) from bone marrow (BM) blood samples of the CML patients were isolated and BM CD34+/lin- cells were selected using a Diamond CD34 Isolation kit and an autoMACS Pro separator (Miltenyi Biotec, Bologna, Italy) according to the manufacturer’s instructions (Miltenyi Biotec). Method details were described in “http://dx.doi.org/10.17504/protocols.io.yncfvaw (assessed on 19 July 2019)” and in our previous study [18].

### 2.3. CML Cell Line

An aliquot of the leukemic cell line KCL22 was kindly provided by Papa Giovanni XXIII Hospital (Bergamo, Italy).

### 2.4. Breakpoint Coordinates

*BCR-ABL1* BPs were characterized through target enrichment and NGS on DNA from patients’ CD34+/lin- cells and the cell line, using a SureSelectQXT Target Enrichment custom panel (Agilent Technologies SpA, Milano (MI), Italy) and paired-end sequencing technologies (Hiseq2500, Illumina, IGA technology services s.r.l., Udine (UD), Italy). Raw reads were aligned to the human chromosome 9 and 22 sequences (GRCh38/hg38) using a BWA-MEM algorithm with default parameters. PCR duplicates were removed by means of the Picard MarkDuplicates tool version:2.23.9, “http://broadinstitute.github.io/picard/ (accessed on 1 May 2022)”. Structural variations (SVs) identification and BPs refinement were performed using DELLY software (version 0.8.7). All the breakpoints’ coordinates were submitted to GenBank (BankIt2713948: OR137812–OR137913).

## 3. Results

### NGS and DELLY Analysis

Target enrichment and NGS characterization of 82 CML patients and one cell line showed peculiar BP DNA coordinates. DELLY analysis and visual inspection with the Integrative Genomics Viewer (IGV) identified a total number of 51556 SVs, among which 3248 were translocation events, in the 83 samples. We identified both Ph and reciprocal der9 chromosomes in 53 samples (106 translocations), whereas Ph alone was identified in 26 samples and der9 in four (Figure 1), resulting in 136 total translocation rearrangements and 272 BPs in total.

DELLY reported 102 of 136 BP junction consensus sequences: 86 were characterized by micro-homologous regions (ranging from one to twenty nucleotides) and two BPs showed non-template insertion sequences of one base at the joined ends of the rearrangements (88%). The remaining 14 junction sequences were blunt (12%) (Table 1 and Appendix A).

Based on the BP coordinates, all samples showed a loss of genomic material at the BP regions either for chr9 or chr22, except for one case.

BPs on chromosome 22 spanned into a genomic region of almost 4.65 Mb (chr22:23289283-27941711). With the exception of one BP located 4.62 Mb distally to *BCR* (belonging to a der9 chromosome) and one BP mapping in the region between M-bcr and μ-bcr, all BPs clustered within the M-bcr boundaries, with most of them mapping in the introns 13, 14, and 15 of the *BCR* gene (NM_021574.3), and only seven BPs in exons (one in exon 13, three in exon 14, and three in exon 15). No BPs were identified inside the m-bcr and μ-bcr. Almost all the BPs on chromosome 9 (130 of 136) mapped inside of intron 1 of the *ABL1* gene (NM_007313.3) in a region of 137.7 Kbs (chr9:130715146-130852871). The remaining six BPs were located proximally to the *ABL1* gene (chr9:130662791-130710869), with one of them in 9p11.2 (chr9:41102675). With the exception of two, all the BPs were embedded into the “*ABL1*” enriched region.

Additionally, 36% of the total BPs (98/272) mapped into a repetitive element (Figure 1). The most represented classes were SINE (43.9%) and LINE (35.7%), followed by DNA transposons, LTR, and simple repeats (20.4% in total) (Figure 1).

Due to the BPs clustering in the M-bcr and *ABL1* regions, we evaluated the genomic content of these intervals of ~9 kbs and ~154 kbs, respectively. Both regions were characterized by the predominant presence of LINEs and SINEs. In detail, based on the UCSC’s RepeatMasker annotation, the *ABL1* region contains 101 LINE elements and 75 of them are L1M* type (Figure 2). Likewise, in the small M-bcr region, five LINEs are annotated and three of them belong to two different L1M* subfamilies (Figure 2).

The screening of the random intervals (RIs, see Materials and Methods) confirmed an enrichment of LINEs as well as SINEs in the regions of BP clustering in our samples. We paid more attention to the LINE elements due to the highest level of enrichment (Table 1). In detail, considering a FC value greater than two, the *ABL1* region was characterized by the enrichment of 13 different LINEs: HAL1ME, L1MCc, L1MEc, L1ME1, L1MEf, L1Med, L1MC5, L1P4, L1ME4a, L1MB4, HAL1, L1M5, and L1MC4. With the exception of HAL1ME, L1P4, and HAL1, all elements are included in the L1M subfamily. Similarly, in M-bcr, we revealed an enrichment of three LINE elements; two of them are members of the L1M subfamily (L1MC1 and L1ME), and one of the L2b family. L1M copy number values in the samples’ M-bcr and *ABL1* intervals resulted significantly higher than the corresponding L1M element mean values in RIs (Appendix A).

Pairwise alignment identified 12 DNA blocks ranging from 119 to 250 bps in length and an average sequence identity of 81.5%. In M-bcr, 11 out of 12 sequences mapped closely to a L1ME1 element, while the remaining one was in proximity of a L1MC1 repeat, and along the *ABL1*-enriched region the blocks showed a scattered distribution; like L1M repeat elements, the identified sequence similarity was derived from Alu sequences (Figure 3 and Appendix A).

The finding of an enrichment of the same type of repetitive elements (L1M) in both BP cluster regions potentially involved in the rearrangements is an interesting result described in detail for the first time.

## 4. Discussion

Transposons represent around 54% of the mammalian genome, of which almost 21% is represented by long interspersed nuclear elements (631,64 Mb) [19]. Most of these elements are silent and inactive, but 10–15% of them play an active role in the regulation of gene expression. In this study, we explored the role of transposable elements in the rearrangements of *BCR* and *ABL1*, investigating if a topological feature or a specific sequence close to these genes could promote their fusion. We characterized patient-specific BP sequences at chr9 and chr22 and we found that almost all 82 CML samples had the M-bcr (mostly in introns 13, 14, and 15). The analysis of regions spanning the BPs of ~9 kbs and ~154 kbs for chr22 and chr9, respectively, showed a predominant presence of SINEs and LINEs. Indications that Alu elements may have a role in the generation of *BCR-ABL* have already been reported [20]. With a fine characterization of the breakpoints mapping, we narrowed the BPs regions involved in the t(9;22) in CML, and we assessed a high presence of LINE1 elements belonging to the L1M subfamily: 10 different L1M elements in the *ABL1* cluster region (L1MCc, L1MEc, L1ME1, L1MEf, L1Med, L1MC5, L1ME4a, L1MB4, L1M5, and L1MC4) and two L1M elements in the M-bcr cluster region (L1MC1 and L1ME). A previous study on a big cohort of samples was not able to identify consensus sequences around breakpoints but they analyzed only a small region across breakpoints [21], while in our study we analyzed a bigger region spanning all the breakpoints of our cohort.

According to the literature, a LINE1 insertion can affect the genome, epigenome, and the whole transcriptome of cells. Studies revealed that LINE1s increase gene mobilization leading to the formation of chimeric genes generating new transcripts that could rise to chimeric proteins. An in silico study found 988 genes with LINE1 insertions that could generate chimeric transcripts, of which twenty have been associated with cancer [22].

The involvement of LINE1s in both gene deletion and chromosomal translocations was demonstrated by Rodriguez-Martinez and colleagues in an esophageal adenocarcinoma and head-and-neck and colorectal cancers. Authors reported that the aberrant L1 integrations can delete large regions of chromosomes, leading to the removal of tumor-suppressor genes and inducing complex translocations and large-scale duplications [23].

Repetitive elements are known to be heavily methylated in normal somatic tissues, but their methylation status is to a lesser extent in malignant tissues, driving the global genomic hypomethylation [24]. The cancer genome is frequently characterized by promoter hypomethylation of specific genes with an overall decrease at the level of five-methylcytosine. This hypomethylation affecting repeat sequences and transposable elements results in chromosomal instability and mutation events [25]. Previous studies demonstrated that hypomethylation of LINE1 promoters and other transposable elements appear at early stages of CML development. Roman-Gomez et al. hypothesized that destabilization of repetitive sequences (i.e., L1 hypomethylation) could be one of such mechanisms employed by *BCR-ABL* to generate genomic instability in the malignant cell, suggesting that repetitive DNA hypomethylation is closely associated with CML progression [26]. In addition, hypomethylation is also associated with open chromatin compartments, which turn out to be easily accessible. Engreitz et al. reported that translocations whose partners lie in the open chromatin regions are more significantly proximal than translocations with one or both partners in the closed compartments. By means of Hi-C experiments using a karyotypically normal lymphoblastoid cell line (GM06990), they demonstrated a significant contact frequency between the BCR and ABL loci [27]. Considering all these data, we could speculate that the higher density of hypomethylated L1M in these regions might be responsible for their nuclear proximity and open chromatin structure, thus allowing more frequent chromosomal rearrangements.

## 5. Conclusions

Our data highlighted the high distribution of L1M in *BCR* and *ABL1* gene regions, suggesting that L1M elements could be potential drivers of the t(9;22) translocation leading to the generation of the *BCR-ABL1* gene and the consequent expression of the BCR-ABL1 active tyrosine kinase chimeric protein. Although it is not possible to confirm that L1M elements work as main actors in mediating the t(9;22) rearrangement, we can argue for a synergic role of theirs with the Alu elements, which can trigger the translocation rearrangement in association with the tendency of *BCR* and *ABL* loci to be spatially proximal [28].

## Figures and Tables

**Figure 1 genes-14-01351-f001:**
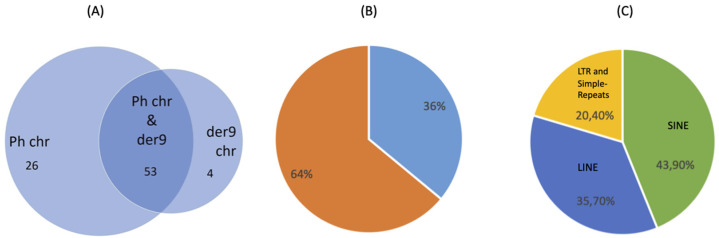
(**A**) Venn diagram shows the number of samples containing the Ph chromosome and/or the der9 chromosome. The overlapping area corresponds to the samples for which both chromosomes have been identified. (**B**) Graph represents the percentage of BPs mapped in repetitive elements (36%) and in non-repetitive elements (64%). (**C**) Graph indicates the distribution of repetitive elements in the BP cluster regions on chr22 and chr9.

**Figure 2 genes-14-01351-f002:**
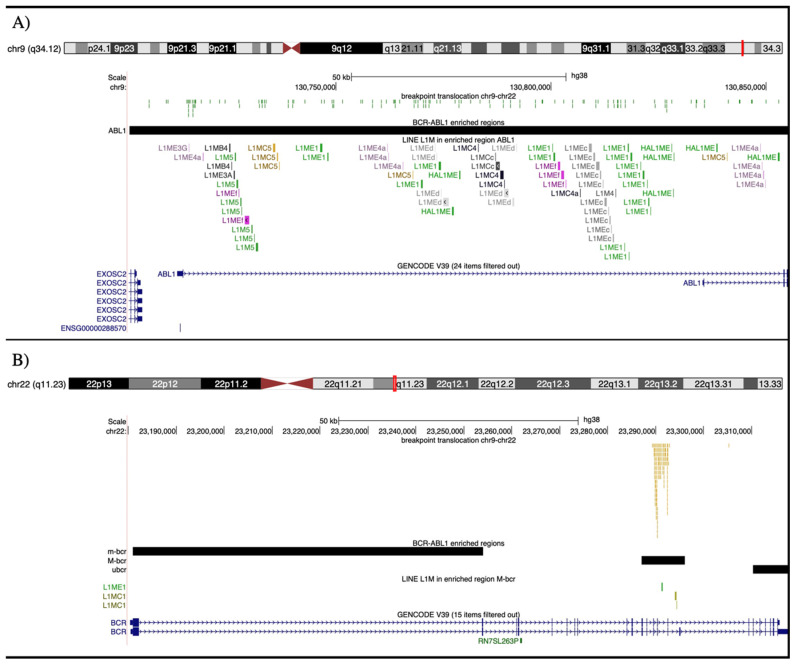
Chromosome 9 ideogram with the corresponding UCSC GENCODE track and three custom tracks reporting the breakpoint localization in patients (green thick), the ABL1 region (black block), and the L1M repetitive element distribution (each type with a different color) (panel (**A**)); Chromosome 22 ideogram with the corresponding UCSC GENCODE track and three custom tracks reporting the breakpoint localization in patients (yellow thick), the m-bcr/M-bcr/μ-bcr regions (black blocks), and the L1M repetitive element distribution (each type with a different color) (panel (**B**)).

**Figure 3 genes-14-01351-f003:**
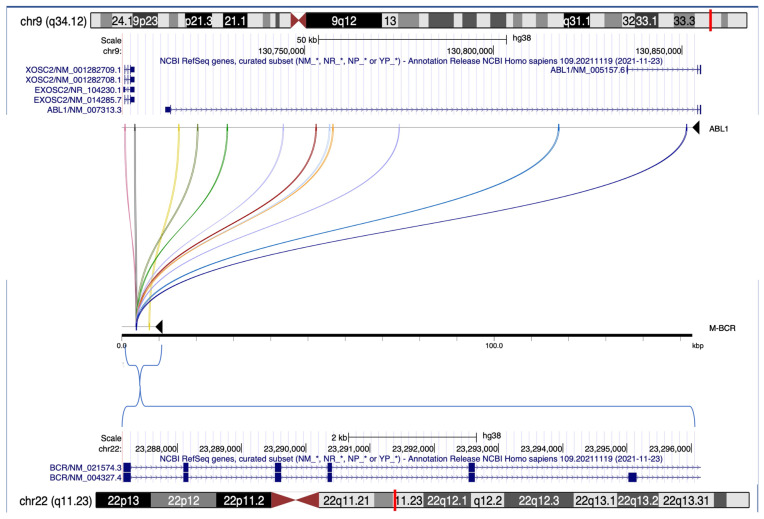
**Pairwise alignment of the breakpoint cluster regions ABL1 and M-bcr.** On the top and the bottom are reported the chromosome 9 and 22 ideograms with the corresponding NCBI RefSeq genes content, respectively. In the middle, pairwise alignment shows the connection of the genomic blocks with a sequence similarity in ABL1 (black line on the top) and M-bcr (black line on the bottom) by means of colored arches.

**Table 1 genes-14-01351-t001:** **Summary information associated with the characterized translocations**. The table reports the total number of translocations identified in the cohort and the number of consensus junction sequences identified with DELLY software. In detail, we have reported the percentage of consensus junction sequences characterized by micro-homologous regions or blunt ends at junction sites, as well as the distribution of BPs on chr9 and chr22, the percentage of those mapping in repetitive elements, and the type of repetitive elements in which they map. Finally, enriched L1M LINE elements in ABL1 and M-bcr regions and their corresponding fold change values are indicated.

**Number of translocations**	136			
**Number of BP consensus sequences**	102			
	88% micro-homologies/non-template insertion		12% blunt	
**BPs chr22**	M-bcr	m-bcr	μ-bcr	outside
	134	-	-	2
**BPs chr9**	ABL1 region		outside	
	128		8	
**36% BPs in repetitive elements**	SINE		43.9%	
	LINE		35.7%	
	other		20.4%	
**L1M LINE enrichment (FC^a^ > 2)**		FC ABL1/RI_154 kb_mean		FC M-bcr/RI_9 kb_mean
	L1ME1	9.5		10.5
	L1MCc	12.4		-
	L1MEc	10.6		-
	L1MEf	8.5		-
	L1MEd	6.6		-
	L1MC5	6.4		-
	L1ME4a	4.7		-
	L1MB4	4.5		-
	L1M5	2.6		-
	L1MC4	2.4		-
	L1MC1	-		50.9

## Data Availability

WGS data are available at the NCBI Sequence Read Archive “https://www.ncbi.nlm.nih.gov/sra (accessed on 29 June 2022)” under accession number PRJNA853897.

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
