# Peer review of "Occurrence of L1M Elements in Chromosomal Rearrangements Associated to Chronic Myeloid Leukemia (CML): Insights from Patient-Specific Breakpoints Characterization"

_genes, 2023, doi:10.3390/genes14071351_

Round 1

Reviewer 1 Report

In the manuscript "Occurrence of L1M elements in chromosomal rearrangements 2 associated to Chronic Myeloid Leukemia (CML): insights from 3 patient-specific breakpoints characterization," L’Abbate et al., showed that L1M elements could be potential drivers of t(9;22) translocation leading to the generation of BCR-ABL1 chimeric gene and the expression of BCR-ABL1 active tyrosine kinase chimeric protein responsible for CML. While the results are impressive, there are some important comments that need to be addressed:

Major Comments:

1.       This manuscript lacks detailed baseline clinical characteristics of the 82 chronic phase (CP)-CML patients. The assessment of patients at different stages is not clearly described, and the rationale for selecting CP-CML patients is unclear.

2.       The authors should include information on treatment guidelines, such as the starting dosage of Nilotinib given to the patients. It is also important to mention if there were any instances of non-adherence during the treatment and provide the dosage of second-generation TKIs, if any used for Nilotinib resistance patience. Without this information, readers may struggle to understand the potential treatment resistance.

3.       Additionally, the authors need to include hematological response (CHR, PHR) at the end of 3 months of Nilotinib therapy, cytogenetic response (CCyR, PCyR) at 6 months after Nilotinib initiation, and molecular response (MMR, CMR) at 12 months after drug initiation for all 82 CML patients.

4.       Furthermore, the authors did not characterize the BCR-ABL1 fusion status (b3a2 or b2a2) in the tested patients with CML. Resistance is often categorized as BCR-ABL dependent or BCR-ABL independent. It would be ideal to verify the involvement of BCR-ABL1 in drug resistance, such as identifying any mutations in the BCR-ABL1 gene that could cause poor drug binding in the ATP region.

5.       The authors should provide Sokal score or Eutos score for the 82 patients prior to Nilotinib exposure, as well as Sokal score or EUTOS score for the 82 CML patients who mostly achieved MMR or CMR. It is important to understand the distribution of high Sokal scores, intermediate Sokal scores, and low Sokal scores among the patients undergoing therapy.

6.       All identified breakpoints in the BCR and ABL1 genes should be deposited in GenBank (https://www.ncbi.nlm.nih.gov/genbank/) and the accession numbers should be provided in the manuscript.  

Author Response

Reviewer1

Comments and Suggestions for Authors

In the manuscript "Occurrence of L1M elements in chromosomal rearrangements 2 associated to Chronic Myeloid Leukemia (CML): insights from 3 patient-specific breakpoints characterization," L’Abbate et al., showed that L1M elements could be potential drivers of t(9;22) translocation leading to the generation of BCR-ABL1 chimeric gene and the expression of BCR-ABL1 active tyrosine kinase chimeric protein responsible for CML. While the results are impressive, there are some important comments that need to be addressed:

Major Comments:

  1. This manuscript lacks detailed baseline clinical characteristics of the 82 chronic phase (CP)-CML patients. The assessment of patients at different stages is not clearly described, and the rationale for selecting CP-CML patients is unclear.

We really thank the reviewer for this comment, we clarified the rationale for selecting CP-CML patients in the methods section.

“The PhilosoPhi34 study enrolled newly diagnosed Ph+ CML patients in chronic phase (CP-CML), aged ≥18 years, either male or female.”

  1. The authors should include information on treatment guidelines, such as the starting dosage of Nilotinib given to the patients. It is also important to mention if there were any instances of non-adherence during the treatment and provide the dosage of second-generation TKIs, if any used for Nilotinib resistance patience. Without this information, readers may struggle to understand the potential treatment resistance.

We really thank the reviewer for this comment, but we would like to stress out that the purpose of this paper was to study genomic rearrangements at sequence level without highlighting any correlation between genomic and pharmacological treatment. Nevertheless, as indicated in Appendix A, we did report the treatment for all patients that received a standard Nilotinib 300 mg BID therapy.

  1. Additionally, the authors need to include hematological response (CHR, PHR) at the end of 3 months of Nilotinib therapy, cytogenetic response (CCyR, PCyR) at 6 months after Nilotinib initiation, and molecular response (MMR, CMR) at 12 months after drug initiation for all 82 CML patients.

According to the previous comment, we would like to highlight that this study focuses on the molecular characterization of the genomic sequences spanning the chromosomal breakpoint and the DNA samples were analyzed at the diagnosis. Unfortunately, the samples at different stages are not available, so we are not able to answer this criticism.

  1. Furthermore, the authors did not characterize the BCR-ABL1 fusion status (b3a2 or b2a2) in the tested patients with CML. Resistance is often categorized as BCR-ABL dependent or BCR-ABL independent. It would be ideal to verify the involvement of BCR-ABL1 in drug resistance, such as identifying any mutations in the BCR-ABL1 gene that could cause poor drug binding in the ATP region.

We thank the reviewer for this comment, but still, it is in the same direction of the previous comments thus posing question far from the purpose of the paper as it is. Unfortunately, we were not able to perform transcriptomic analysis, so we are not able to answer this question.

BCR-ABL1 fusion status can be obtained only from the analysis of mRNA amplicon length, that is not the aim of this work and requires the analysis of RNA material that was not available for this study.

  1. The authors should provide Sokal score or Eutos score for the 82 patients prior to Nilotinib exposure, as well as Sokal score or EUTOS score for the 82 CML patients who mostly achieved MMR or CMR. It is important to understand the distribution of high Sokal scores, intermediate Sokal scores, and low Sokal scores among the patients undergoing therapy.

The molecular study we performed was focused on the chromosomal sequence at the breakpoint of 82 patients in order to understand and analyze the potential presence of repetitive elements or other specific sequences at the basis of a reciprocal translocation between chromosome 22 and 9. Results of this study could be of interest for the prediction of potential translocation hotspots and does not have any correlation with any clinical evaluation, diagnosis or prognosis.

  1. All identified breakpoints in the BCR and ABL1 genes should be deposited in GenBank (https://www.ncbi.nlm.nih.gov/genbank/) and the accession numbers should be provided in the manuscript.  

We thank the reviewer for this helpful comment and we now submitted the breakpoints to Genbank and add this information into the paper in the “Breakpoint coordinates” section.

Reviewer 2 Report

In this paper L'Abbate and colleagues explore the occurrence of L1M elements in 82 CML patients and 1 cell line, by NGS strategies they analyzed determined regions of the genome and discussed the potential role of L1M elements as drivers of pathogenic translocations resulting in CML.  

The manuscript is well organized and easily readable, the references are adequate, and no inappropriate self-citations were detected. The described methods allows the results reproducibility, the results and the conclusions are consistent, and no ethical issues were detected. 

as minor comments: 

-Figures: in some figures the sections are indicated using just letters, whereas in other figures employ letters with parenthesis, it must be homogenized  

-Paragraphs are justified in some parts while in others are left aligned it must be homogenized.

-Indents are used in some paragraphs while in others not (e.g., introduction), it must be homogenized.

-All the Latin words (e.g., loci) must be in italics.

-I could not find the following references within the text, that is they are not cited by its designed number or by the principal author…

Trojani et al., 2019 (29)

Pungolino et al., 2018 (30)

Author Response

Reviewer2

In this paper L'Abbate and colleagues explore the occurrence of L1M elements in 82 CML patients and 1 cell line, by NGS strategies they analyzed determined regions of the genome and discussed the potential role of L1M elements as drivers of pathogenic translocations resulting in CML.  

The manuscript is well organized and easily readable, the references are adequate, and no inappropriate self-citations were detected. The described methods allows the results reproducibility, the results and the conclusions are consistent, and no ethical issues were detected. 

as minor comments: 

-Figures: in some figures the sections are indicated using just letters, whereas in other figures employ letters with parenthesis, it must be homogenized  

All the figures have been homogenized.

-Paragraphs are justified in some parts while in others are left aligned it must be homogenized.

We really thank the reviewer's comment, and all the paragraphs are now justified.

-Indents are used in some paragraphs while in others not (e.g., introduction), it must be homogenized.

We really thank the reviewer's comment, and everything has been homogenized.

-All the Latin words (e.g., loci) must be in italics.

We modified the Latin words accordingly.

-I could not find the following references within the text, that is they are not cited by its designed number or by the principal author…

Trojani et al., 2019 (29)

Pungolino et al., 2018 (30)

We thank the reviewer for this comment, but we would like to highlight that both papers are correctly numbered and cited in the Appendix A at the sectionPatient material”, so we did not modify anything considering this point.

Round 2

Reviewer 1 Report

The authors have adequately addressed all the questions raised, providing satisfactory responses. Based on their efforts, I highly recommend that the manuscript be deemed acceptable for publication in the journal Genes.